# Audio-FLAN: An Instruction-Following Dataset for Unified Understanding and Generation of Speech, Music, and Sound

## Abstract

Instruction tuning has generalized well in language and vision, yet audio remains siloed by domain (speech, music, environmental sound) and by task type (understanding vs. generation). We present Audio-FLAN, a large-scale instruction-following corpus that unifies heterogeneous audio sources under a unified instruction schema with *instruction*, *input*, and *output*. It supports both understanding (audio→text) and generation (text/audio/{audio,text} →audio) across speech, music, and general audio. The dataset contains 108.5M instances spanning 23 major and 80 minor tasks drawn from 52 datasets. Instruction tuning on a small subset of Audio-FLAN yields consistent gains on diverse understanding tasks, including zero-shot generalization. We further evaluate the existing generation model and validate Audio-FLAN as an effective benchmark. Hallucination probes inform future data curation and training design. In summary, Audio-FLAN serves as both an effective training resource and a unified, extensible benchmark for instruction-following audio–language models. We release the dataset on HuggingFace.

## 1 Introduction

Instruction tuning has proven to be a simple yet powerful recipe for broad generalization in language and vision: aligning models to follow natural-language instructions across diverse tasks yields strong zero/few-shot transfer beyond pretraining alone (Ouyang et al., 2022; Wei et al., 2021a; Touvron et al., 2023; Achiam et al., 2023). In NLP, FLAN-style multi-task tuning substantially improves zero-shot performance (Wei et al., 2021a), and even relatively small but well-curated instruction datasets can achieve performance comparable to far larger models (Zhou et al., 2024). Multimodal systems further demonstrate that a single unified model can seamlessly handle both understanding and generation tasks (Team, 2024). By contrast, audio research remains fragmented: speech, music, and environmental sound are often studied in isolation, and *understanding* (e.g., recognition, transcription, captioning) is typically decoupled from *generation* (e.g., text to speech, conversion, enhancement), leaving few truly unified audio–language models.

Despite recent advances, unifying audio tasks cannot be accomplished by model capacity alone, but demands dedicated modeling and data design. Learned audio codecs such as SoundStream and EnCodec expose discrete sequences amenable to language-modeling toolchains (Zeghidour et al., 2021; Défossez et al., 2022), and text-conditioned generators, such as AudioLM, MusicLM, AudioGen, Make-An-Audio (Borsos et al., 2023; Agostinelli et al., 2023; Kreuk et al., 2023; Huang et al., 2023), demonstrate high-fidelity speech/music/sound synthesis. Meanwhile, audio–language systems align audio encoders with LLMs for open-ended understanding across speech, music, and environmental audio (Tang et al., 2023; Chu et al., 2024; Kong et al., 2024), and existing research frameworks for instruction-to-audio address only a subset of generation tasks (Yang et al., 2023). Public instruction-following audio–language models that support *both* understanding and generation remain scarce. Most available systems, e.g., Qwen2-Audio (Chu et al., 2024), Step-Audio (Huang et al., 2025), Baichuan-Audio (Li et al., 2025), Kimi-Audio (KimiTeam et al., 2025), primarily target understanding and conversational use rather than instruction-following audio synthesis. Moreover, current community benchmarks, e.g., Dynamic-SUPERB (yu Huang et al., 2024), Dynamic-SUPERB Phase-2 (Huang et al., 2024), MMAU (Sakshi et al., 2024), AIR-Bench (Yang et al., 2024), still

center on understanding evaluation, seldom adopt a unified instruction-following schema across domains and output modalities(yu Huang et al., 2024; Huang et al., 2024; Sakshi et al., 2024; Yang et al., 2024). In short, modeling advances are necessary but insufficient: the field lacks a large-scale, instruction-following dataset and protocol that jointly span understanding and generation over speech, music, and general audio.

We introduce AUDIO-FLAN, a large-scale instruction-following dataset that standardizes heterogeneous audio sources into a single schema for both understanding and generation across speech, music, and general audio. Each JSONL record pairs a natural-language instruction with audio/text inputs and text/audio outputs. Following Self-Instruct, we generate multiple rephrasings of each instruction–input pair so the model remains robust to wording changes and still produces the correct outputs (Wang et al., 2023b). The dataset provides train/dev/test splits and zero-shot (unseen-task) configurations to enable comparable training and evaluation. Empirically, instruction tuning on Qwen2-Audio (Chu et al., 2024) with a small subset of AUDIO-FLAN yields consistent gains on understanding tasks. Evaluation of UNIAUDIO (Yang et al., 2023) on the AUDIO-FLAN test set, under a unified protocol spanning various generation tasks, indicates that AUDIO-FLAN provides a task-aligned, cross-domain benchmark. Hallucination analysis reveals failure cases that inspire future data and training design.

Our contributions are: (i) a *unified instruction schema and corpus* that cover both understanding and generation across speech, music, and general audio; (ii) an *instance-level Self-Instruct* diversification pipeline that varies phrasing and intput/output prefixes while preserving task semantics; (iii) a *benchmark protocol* with standardized train/dev/test splits and seen/unseen configurations plus task-appropriate evaluation interfaces; and (iv) *empirical validation and analysis* showing understanding improvements from instruction tuning, reproducible generation evaluation on multiple tasks, and a hallucination study that surfaces current limits and motivates uncertainty-aware data design.

## 2 AUDIO-FLAN DATASET

### 2.1 TASK TAXONOMY

We organize tasks hierarchically into *major* and *minor* categories across three domains—SPEECH, MUSIC, and AUDIO. Major tasks denote functional families (e.g., *Speech Recognition*, *Audio Event Recognition*, *Music Generation*), while minor tasks instantiate concrete objectives within each family (e.g., *Automatic Speech Recognition* and *Dialect Speech Recognition* under *Speech Recognition*; *Event Recognition* and *Sound Event Sequence Recognition* under *Audio Event Recognition*; *Lyrics-to-song* and *Music Continuation* under *Music Generation*). This *extensible* hierarchy clarifies task scope and specialization: new minor tasks can be attached to existing families, and new major families can be introduced as the field evolves. The taxonomy not only covers common understanding and generation tasks but also explicitly includes underexplored music *time-sequential* reasoning. For example, Beat-level Pitch Estimation and Beat-level Instrument Recognition (under *Single Music Reasoning*) require interpreting musical elements at specific time points, whereas Tempo/Key/Instrument/Emotion Comparison (under *Multiple Music Reasoning*) involves comparing musical features over time. These tasks challenge models to generalize over complex, time-dependent patterns. In total, AUDIO-FLAN spans *23 major* and *80 minor* tasks across speech, music, and audio, providing a coherent, extensible mapping from diverse audio-related tasks to a unified instruction format and enabling the training of general-purpose audio–language models. A full task taxonomy is presented in Table 5 in the Appendix.

### 2.2 UNIFIED TASK TEMPLATE

We adopt a *Unified Task Template (UTT)*, an instruction schema that is agnostic to both task and modality. Let $\{I_i\}$ denote the collection of natural-language *instructions*, where each $I_i$ describes a specific task $i$. For every task $i$, there exist $n_i \geq 1$ *input–output* pairs $\{(X_{t,i}, Y_{t,i})\}_{t=1}^{n_i}$. Once the task set is fixed, each example is represented in JSONL (JSON Lines) with three core fields: **instruction**, **input**, and **output**. The **instruction** is a concise task description that tells the model what input to expect and what type of output to produce. For *understanding* tasks, the **output** is *text*; for *generation* tasks, the **output** is typically *audio*. The **input** can be *audio*, *text*, or a mixture of both, depending on

the task. Formally, given an instruction–input pair $(I_i, X_{t,i})$, a model $M$ is expected to produce the corresponding output $M(I_i, X_{t,i}) = Y_{t,i}$ for $t \in \{1, \ldots, n_i\}$.

We use $<|SOA|>$ to mark the start of audio and $<|EOA|>$ to mark the end of audio; when the input contains multiple values, they are separated by \n. JSONL files also include metadata fields such as uuid, split, task_type, and domain. The complete schema and specific examples are presented in Appendix A.2. During dataset processing, each task's UTT is instantiated into a *Template-Instantiated Record (TIR)* by filling **instruction**, **input**, and **output** with sample-specific values; the subsequent variation stage further diversifies a TIR into a *Varied Instance Record (VIR)* while preserving semantics and schema.

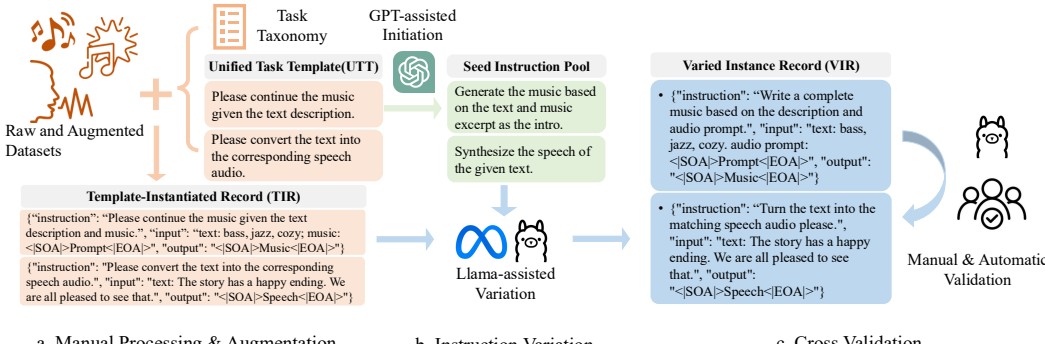

Figure 1: Overview pipeline of Audio-FLAN dataset construction.

## 2.3 DATASET PROCESSING

The overview pipeline of AUDIO-FLAN dataset construction is illustrated in Figure 1. Based on datasets collected from open-source resources or augmentation and the Unified Task Template (UTT) defined above, we materialize sample-level *Template-Instantiated Records (TIRs)* in a unified JSONL schema with three core fields: *instruction*, *input*, and *output*. For *understanding* tasks (audio→text), we turn available annotations into outputs in the TIR format, while normalizing labels (naming/merges/casing) and checking spans/timestamps as needed. For *generation* tasks (text/audio/{audio,text} →audio), we use raw dataset when available; otherwise, we *augment dataset* via controlled transformations—adding noise/reverberation/clipping for enhancement/dereverberation, bandwidth changes for super-resolution, source mixing for separation/extraction—and create alignment/continuation scaffolds for instruction-to-audio synthesis. All resulting examples conform to the UTT schema when instantiated as TIRs.

Each source dataset is partitioned into *train/dev/test*. These splits support (i) instruction-following pre-training or fine-tuning on train, (ii) hyperparameter selection and early stopping on dev, and (iii) final, reportable benchmarking on test. In addition, we provide optional *zero-shot* configurations in which entire minor tasks are tagged as *unseen*, enabling evaluation of cross-task generalization. During processing, each sample fills in its task's UTT as a TIR by setting the **instruction**, **input**, and **output** fields with sample-specific values (setting identifiers such as audio_id and uuid, and adding task/domain metadata). We then check that each JSONL record matches the schema, ensure identifiers are consistent across fields, and remove duplicates across sources. TIR examples are shown in Appendix A.3. The next variation step (described later) turns TIRs into *Varied Instance Records (VIRs)* while keeping the meaning and format unchanged.

## 2.4 RECORD VARIATION AND VALIDATION

Building on the Unified Task Template (UTT) and its Template-Instantiated Records (TIRs), we generate Varied Instance Records (VIRs) via a *self-instruct* strategy (Wang et al., 2023b) tailored to audio-language data. Concretely, we first use GPT-4O to initialize a small, task-level *seed pool* of semantically consistent instruction phrasings that exemplify permissible styles (e.g., imperative vs.

interrogative, concise vs. descriptive). Then, for each TIR, we prompt LLAMA-3.1-70B-INSTRUCT [1] to produce a VIR under strict constraints that preserve the instance intent, the UTT schema (field names and types), and all identifiers (e.g., `audio_id`, `uuid`); the model may paraphrase the **instruction** and lightly rewrite textual prefixes in **input/output** for clarity, but must not alter labels, spans, or timestamps for understanding tasks, and must leave audio outputs unchanged for generation tasks (only their textual context is varied).

We validate every VIR with automatic checks (JSONL schema conformance, identifier consistency, audio markers `<|SOA|>`/`<|EOA|>`, etc.) followed by stratified manual spot-checks per minor task family to ensure semantic equivalence and label/timestamp fidelity. This self-instruct pipeline—seeding with GPT-4O, constrained rewriting with LLAMA, and rigorous validation—proves a practical and reliable way to obtain stylistically diverse yet schema-faithful VIRs. Variation prompt and VIR examples are provided in the Appendix A.4 and A.5. We release the complete set of finally processed VIR JSONL for all tasks on HuggingFace.

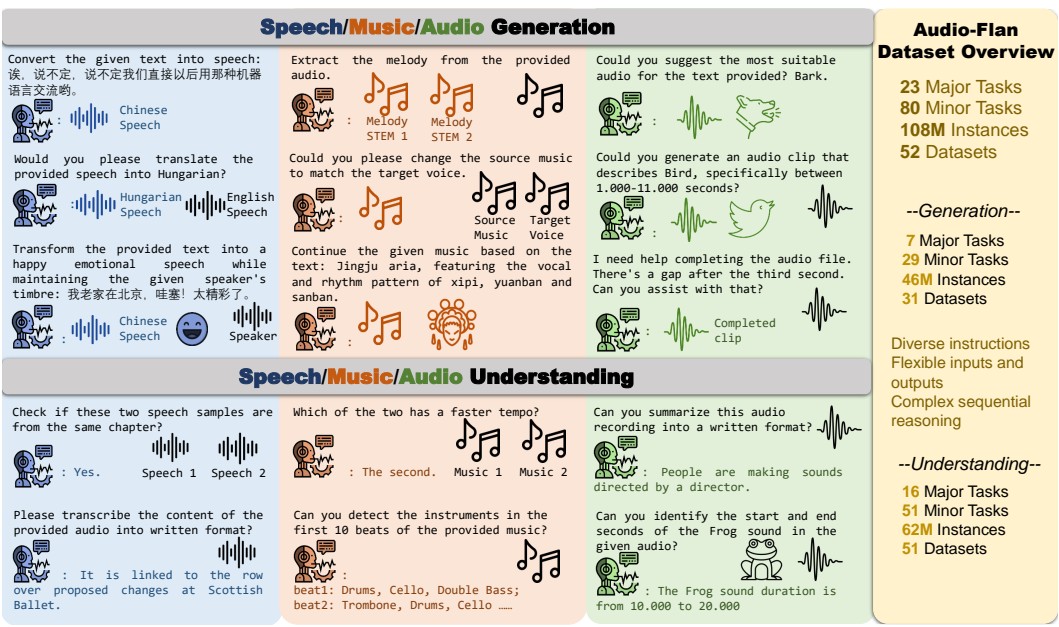

Figure 2: Overview of Audio-FLAN dataset. Examples of **speech**, **music**, and **audio** understanding and generation tasks are illustrated.

## 2.5 STATISTICS

**Instances Statistics**   Figure 2 presents AUDIO-FLAN dataset overview, comprising *108.5M* instruction–instance pairs covering *23 major* and *80 minor* tasks aggregated from *52* source datasets.[2] The corpus is split into *generation* (46.06M; 7 major/29 minor; 31 datasets) and *understanding* (62.44M; 16 major/51 minor; 51 datasets). Tasks span speech, music, and general audio with flexible input–output pairings (text↔audio, audio↔audio, and mixed) and include time-sequential reasoning. By domain, speech contributes 100.42M instances, music 2.17M, and general audio 5.91M. While speech is over-represented, AUDIO-FLAN provides substantive cross-domain coverage under a unified instruction format, enabling zero-shot evaluation and instruction tuning for unified audio–language models. Downstream users may apply reweighting or sampling to mitigate the acknowledged imbalance as described in Section 2.6. Detailed instance statistics of each major task are listed in Table 6.

**Attributes Distribution**   Figure 3 illustrates the breadth of attributes covered by Audio-FLAN dataset across speech, music, and audio domains. In panel (a) on speech, content (35.5%) and

---

[1]https://huggingface.co/meta-llama/Llama-3.1-70B-Instruct

[2]A single source dataset may contribute to multiple task families; the dataset count is deduplicated.

language (32.1%) are most common, alongside labels such as gender, age, and dialect. Panel (b) on music highlights vocals (19.4%), instrumental (17.6%), and timbre (12.9%), together with melody, pitch, and ethnomusicology descriptors that reflect structural and cultural diversity. Panel (c) on general audio shows scene (33.4%), event (22.2%), and speech (20.3%) as leading categories, plus a heterogeneous "other" group that includes quality-related tags such as super-resolution. The observed attribute distribution demonstrates broad coverage of audio properties and diverse task categories, enabling unified modeling and promoting robust generalization in real-world audio–language settings.

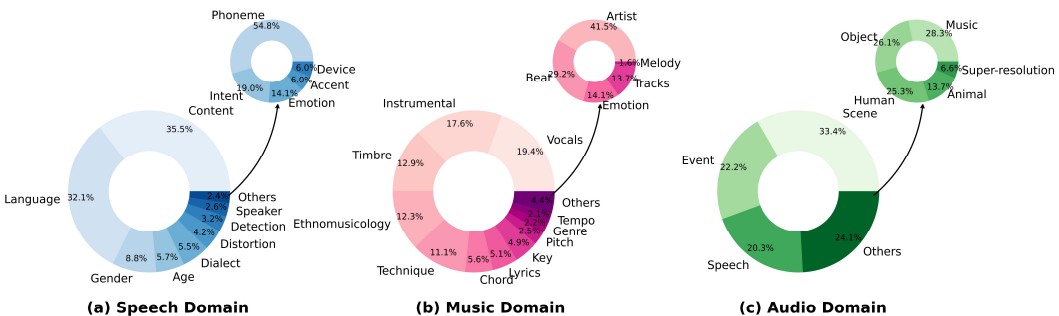

Figure 3: Distribution of attributes in Audio-FLAN across (a) speech, (b) music, and (c) audio domains.

### 2.6 KNOWN LIMITATIONS AND SUGGESTED USAGE

AUDIO-FLAN is intentionally broad but not balanced: speech contributes most instances, while music and general audio are less represented. At the task level, understanding is more common than generation. This skew reflects data availability and licensing realities: large speech corpora are easier to obtain, whereas general audio and especially music are scarcer and subject to tighter copyright constraints. The rich attribute diversity within speech further enables many understanding tasks, reinforcing this imbalance. In practice, naive training may overfit to speech conventions, inflate aggregate scores (dominated by speech), and understate performance on minority domains and tasks.

To obtain fairer training and evaluation under the known speech–heavy imbalance, we suggest: (i) *balanced sampling* over domains/tasks using a temperature schedule $p_i \propto n_i^\alpha$, $\alpha \in [0, 1]$ (smaller $\alpha$ upweights minority groups, e.g., $\alpha=0$ uniform, $\alpha\approx0.5$ square–root), a practice common in multilingual/multi-domain training (Arivazhagan et al., 2019; Conneau et al., 2020); (ii) *loss reweighting* with inverse-frequency or "effective number" weights, or focal losses (Cui et al., 2019; Ren et al., 2018; Lin et al., 2017); (iii) *balanced mini-batches* that enforce per-domain/task quotas (Buda et al., 2018); (iv) a *two-stage schedule*—domain-specific warmup then joint instruction tuning—akin to curriculum/domain-adaptive pretraining (Bengio et al., 2009; Gururangan et al., 2020); and (v) *stratified model selection and reporting* with per-domain validation, macro-averages that do not track dataset size, and clear separation of seen-task vs. zero-shot (unseen-task) results, following best practices for robustness and generalization reporting (Koh et al., 2021; Mitchell et al., 2019; Wei et al., 2021b). The dataset includes domain/task tags to enable these protocols, and we are expanding underrepresented domains and generation tasks coverage in future releases.

## 3 EXPERIMENTS

### 3.1 EXPERIMENTAL SETUP

AUDIO-FLAN is a large-scale instruction-following dataset for unified audio–language modeling, covering both understanding and generation across speech, music, and general audio. Rather than replicating existing benchmarks, we assess the utility and generalization of Audio-FLAN via three experiments: (i) *instruction-tuned understanding*, (ii) *generation-side evaluation*, and (iii) *zero-shot evaluation*. Finally, we perform a hallucination analysis of the instruction-tuned model to characterize failure modes and inform subsequent data curation and training design.

- **Instruction-tuned understanding.** We fine-tune QWEN2-AUDIO-7B-INSTRUCT[3] (Chu et al., 2024) for 2 epochs with AdamW (Loshchilov & Hutter, 2017) ($lr = 1e - 4$) and LoRA (Hu et al., 2022) ($r = 64$, $\alpha = 192$) adapter under a global batch size of 32 using 8 H800 GPUs on 10% of the AUDIO-FLAN training split restricted to *seen* tasks. Sampling is stratified to approximately balance domains and task types within this subset, and *no* unseen-task data is used during tuning.
- **Generation-side evaluation.** To probe the generation side under a unified protocol, we therefore evaluate UNIAUDIO (Yang et al., 2023)—an open-source model that supports 11 audio generation tasks with multimodal conditioning (not strictly instruction-following)—on the AUDIO-FLAN test set. This provides a practical proxy to assess our dataset's task coverage as a benchmark for instruction-to-audio evaluation.
- **Zero-shot evaluation.** To assess generalization beyond in-domain fitting, we hold out a suite of taxonomy-level tasks as *unseen* and exclude all data from these tasks during instruction tuning. We then evaluate the instruction-tuned model directly on the held-out set. This protocol probes instruction adherence and cross-domain transfer under zero-shot conditions;

### 3.2 EVALUATION METHOD

Because instruction-following outputs are free-form for understanding tasks, we adopt a *large language model* (LLM)–based normalization pipeline following Dynamic-SUPERB Phase-2 (yu Huang et al., 2024): a LLM-based extractor converts raw responses generated from the model into task-specific schemas (e.g., categorical *label*, or *timestamp* tuples for temporal outputs). We log raw outputs, extracted answers, and extraction failures (e.g., empty or ill-formatted). To ensure correctness and fairness, three annotators specializing in speech, music, and general sound conduct expert-guided review against the original instruction and reference, followed by cross-domain secondary review; disagreements are resolved by discussion. *All metrics are computed only on verified extractions*, and outputs marked *unparseable* (empty, off-topic, or format-violating) are *excluded* from scoring. We then report metrics grouped by task type—understanding and generation—as detailed below.

- **Understanding metrics.** *Accuracy (ACC)*: proportion of samples with exactly correct predictions (higher is better), used for classification tasks. *Group Accuracy*: per-sample accuracy averaged over segments or multi-labels and then averaged over the dataset (higher is better), used for segmental or multi-label settings. *BLEU* (Bilingual Evaluation Understudy): $n$-gram precision with brevity penalty for text generation (higher is better), used for captioning/translation quality. *ROUGE-L* (Recall-Oriented Understudy for Gisting Evaluation, longest common subsequence): overlap of the longest common subsequence between prediction and reference, capturing recall-oriented content coverage (higher is better). *Time Overlap Rate (TOR)*: temporal intersection-over-union between predicted and reference events/segments along the time axis (range $[0, 1]$; higher indicates better temporal localization).
- **Generation metrics.** *Word Error Rate (WER)*: Levenshtein distance between hypothesis and reference words normalized by reference length (lower is better), quantifying intelligibility for text-to-speech and voice conversion transcripts. In our evaluation, WER is computed using the HuBERT-Large (Hsu et al., 2021) model fine-tuned on LibriSpeech 960h as the ASR system, following the protocol in Wang et al. (2023a). *Cosine Similarity of speaker embeddings (COS-SIM)*: cosine similarity (range $[-1, 1]$; higher indicates better speaker similarity) between speaker embeddings extracted by the WavLM-TDNN model (Chen et al., 2022) from generated and reference speech, used for TTS/VC timbre preservation. *Signal-to-Noise Ratio (SNR)*: ratio of signal power to noise power in decibels (dB; higher is better), reflecting denoising/separation effectiveness. *PESQ* (Perceptual Evaluation of Speech Quality): perceptual speech-quality score reported in narrowband (NB) and wideband (WB) variants (higher is better), used for speech enhancement and target speech extraction. *STOI* (Short-Time Objective Intelligibility): correlation-based intelligibility index in $[0, 1]$ (higher is better), measuring short-time intelligibility for enhancement/extraction outputs.

### 3.3 INSTRUCTION TUNING ON UNDERSTANDING TASKS

Table 1 shows that instruction tuning on AUDIO-FLAN yields *consistent gains* across speech, music, and general sound, with the largest effects on music reasoning and general audio classification. In music, fine-grained/comparative tasks see substantial jumps (e.g., *Instrument Comparison* $0.03 \rightarrow$

---

[3]https://huggingface.co/Qwen/Qwen2-Audio-7B-Instruct

Table 1: Results before and after instruction-tuning with Audio-FLAN understanding tasks.

| Domain | Task | Before FT | After FT | Metric↑ |
|--------|------|-----------|----------|---------|
| **Seen Tasks** | | | | |
| Music | Key Comparison | 0.040 | 0.205 | ACC |
| | Tempo Comparison | 0.050 | 0.375 | ACC |
| | Instrument Comparison | 0.030 | 0.820 | ACC |
| | Key Detection | 0.080 | 0.125 | ACC |
| | Instrumental Technique Comparison | 0.222 | 0.414 | ACC |
| | Genre Classification | 0.048 | 0.626 | ACC |
| | Instrumental Technique Classification | 0.060 | 0.660 | ACC |
| | Emotion Classification | 0.210 | 0.330 | ACC |
| | Instrument Classification | 0.353 | 0.735 | ACC |
| | Music Tagging | 0.250 | 0.750 | ACC |
| | Music Caption | 0.226/0.054/0.165 | 0.240/0.056/0.218 | BLEU-1/BLEU-4/ROUGE-L |
| Speech | Speaker Verification | / | 0.333 | ACC |
| | Spoken Paragraph Recognition | 0.180 | 0.390 | ACC |
| | Speech-to-text Translation | 0.088/0.035/0.161 | 0.201/0.082/0.333 | BLEU-1/BLEU-4/ROUGE-L |
| | Speech Caption | 0.047/0.000/0.052 | 0.114/0.023/0.083 | BLEU-1/BLEU-4/ROUGE-L |
| Audio | Sound Event Recognition | 0.140 | 0.790 | ACC |
| | Speech, Silence, Music, and Noise Classification | 0.061 | 0.889 | ACC |
| | Vocoder Type Classification | / | / | ACC |
| | Sound Event Detection | 0.029 | 0.305 | TOR |
| | Sound Event Sequence Recognition | 0.027/0.006/0.071 | 0.112/0.041/0.155 | BLEU-1/BLEU-4/ROUGE-L |
| **Unseen Tasks** | | | | |
| Music | Scale Recognition | 0.021 | 0.310 | ACC |
| | Chord Estimation | 0.011 | 0.108 | ACC |
| | Chord Recognition | 0.021 | 0.286 | ACC |
| | Progression Extraction | / | 0.125 | ACC |
| | Beat-level Instrument Recognition | / | 0.414 | ACC |
| | Beat-level Pitch Estimation | / | 0.626 | ACC |
| Speech | Deepfake Detection | / | 0.660 | ACC |
| Audio | Deepfake Audio Detection | 0.210 | 0.330 | ACC |

0.82 ACC; *Instrument/Genre* classification both improve to strong accuracies), indicating better adherence to instruction phrasing beyond standard speech tasks. In general sound, core recognition and coarse scene parsing improve markedly (e.g., *Sound Event Recognition* $0.14 \rightarrow 0.79$ ACC; *Speech/Silence/Music/Noise* rises to $0.889$ ACC), and temporal localization is noticeably stronger (Time Overlap Rate increases for *Sound Event Detection*). Speech understanding also benefits: *Spoken Paragraph Recognition* improves (from $0.18$ to $0.39$ ACC), and free-form outputs become more on-task, with *Speech Caption* and *s*howing steady gains, while *Speech-to-text Translation* improves by roughly $+0.11$ BLEU-1 and $+0.17$ ROUGE-L.

Notably, some tasks that were previously unparseable and whose instructions were not understood by models at all become solvable after tuning ("/" before FT), such as *Speaker Verification* (now reported at $0.333$ ACC), evidencing improved instruction following. Remaining challenges persist in pitch-related and affective understanding (e.g., *Key Detection*, *Emotion Classification*), where gains are smaller. Overall, instruction-tuning with AUDIO-FLAN delivers broad, often large improvements on classification and temporal localization, unlocks tasks the base model could not handle, and brings measurable—though smaller—gains on open-ended generation. These results demonstrate that AUDIO-FLAN is effective instruction-tuning data: it substantially improves instruction-following ability, strengthens audio-grounded reasoning, and yields gains across classification, temporal localization, and open-ended generation.

## 3.4 GENERATION MODEL EVALUATION

We benchmark UNIAUDIO on four instruction-conditioned generation tasks in AUDIO-FLAN: text-to-speech (TTS), voice conversion (VC), speech enhancement (SE), and target speech extraction (TSE). Music generation is not reported (the model failed to produce valid outputs), and for general non-speech sound the model exposes only audio editing, which is outside our current benchmark. Results are summarized in Table 2.

Across TTS corpora, intelligibility is highly domain-sensitive: read speech (e.g., LibriTTS-R, VCTK) yields substantially lower word-error rates than conversational sets (e.g., Common Voice), whereas

Table 2: Results of generation tasks test on UniAudio with Audio-FLAN datasets.

| Task | Test Data | WER↓ | COS↑ | MCD↓ | FAD↓ | SNR↑ | PESQ$_{WB}$↑ | PESQ$_{NB}$↑ | STOI↑ |
|------|-----------|------|------|------|------|------|---------|---------|-------|
| TTS | LibriTTS-R | 0.168 | 0.898 | 7.187 | 0.688 | – | – | – | – |
|  | VCTK | 0.167 | 0.824 | 5.390 | 0.779 | – | – | – | – |
|  | common-voice | 0.490 | 0.828 | 5.396 | 0.734 | – | – | – | – |
| VC | ESD | 0.204 | 0.819 | 5.682 | 0.794 | – | – | – | – |
| SE | DNS-for-Denoising | – | – | – | – | 3.621 | 1.699 | 2.111 | 0.634 |
| TSE | LibriMix | – | – | – | – | 0.687 | 1.202 | 1.367 | 0.402 |

speaker-embedding cosine similarity remains consistently high. This indicates robust timbre transfer even when transcripts degrade. VC on ESD follows a similar pattern: speaker similarity is stable, but spectral/distortion and distributional indicators (MCD, FAD) reveal noticeable artifacts—i.e., reasonable identity preservation with quality gaps under domain mismatch. SE on DNS delivers moderate gains in perceptual quality and intelligibility, while TSE on LibriMix remains notably challenging, with clearly lower PESQ/STOI than SE, reflecting the difficulty of instruction-conditioned extraction from realistic mixtures.

These trends match well-known effects: conversational speech departs from the training distribution of many TTS/VC systems (spontaneous speaking style, accents, background noise), hurting intelligibility more than identity; enhancement improves audibility but cannot fully recover quality under heavy noise; and single-target extraction from mixtures is intrinsically harder than denoising. That AUDIO-FLAN surfaces these contrasts across tasks and domains—under a unified schema and common metrics for intelligibility, similarity, and perceptual quality—supports the soundness of the evaluation. In short, AUDIO-FLAN provides task-aligned, cross-domain test sets that are compatible with existing instruction-conditioned models. This validates AUDIO-FLAN as an effective benchmark for instruction-conditioned generation, and its unified design makes it straightforward to extend coverage as stronger music and sound-generation models emerge.

## 3.5 ZERO-SHOT EVALUATION

Zero-shot results are shown at the bottom of Table 1. Before tuning, the model shows negligible performance on unseen, advanced MIR tasks (ACC $\leq 0.021$) and often produces outputs that cannot be parsed for scoring. While, after tuning on *seen* tasks only, it attains non-trivial accuracy on *Scale Recognition* (0.310) and *Chord Recognition* (0.286), and becomes able to handle previously unparseable beat-level tasks: *Beat-level Instrument Recognition* (0.414) and *Beat-level Pitch Estimation* (0.626). Even where absolute accuracy remains modest (e.g., *Chord Estimation* 0.108), gains over near-zero baselines suggest the model has learned to follow instructions and reason from audio rather than guess. On spoofing tasks that were not used for tuning, the model moves from unparseable or weak baselines to meaningful predictions: *Speech Deepfake Detection* improves from unparseable to 0.660 ACC, and *Audio Deepfake Detection* rises from 0.210 to 0.330 ACC.

These gains, achieved *without* any task-specific supervision for the held-out unseen tasks, show that instruction tuning on AUDIO-FLAN's diverse, instruction-aligned coverage strengthens cross-task generalization. Specifically, the model learns to follow novel instructions, produce valid outputs on previously unsolved tasks, and transfer across domains (music ↔ speech/general audio). Although absolute performance still leaves room for improvement on fine-grained music-theory tasks and difficult detection settings, the consistent improvements substantiate AUDIO-FLAN's effectiveness for enhancing zero-shot capability.

## 3.6 HALLUCINATION ANALYSIS

Hallucination is pervasive in large models. Following the protocol of Kuan et al. (2024), we probe our instruction-tuned QWEN2-AUDIO with both *discriminative* and *generative* [4] tests. In the discriminative setting, we adopt the Polling-based Object Probing Evaluation (POPE) Li et al. (2023), adapted to audio: the model answers binary object-presence questions constructed with

---

[4]Here, "generative" denotes *text* generation used to diagnose hallucination and is *distinct* from AUDIO-FLAN's instruction-to-*audio* generation tasks.

*Random*, *Popular*, and *Adversarial* negatives, and we report accuracy, precision, recall, F1, and the proportion of YES responses ("yes rate"). In the generative setting, we compute the instance-level hallucination score (ECHO-I), the sentence-level hallucination score (ECHO-S), and coverage (Cov), which quantifies how much of the ground-truth audio caption is preserved in the model's output.

Table 3: Results of discriminative tasks in audio captioning on instruction-tuned Qwen2-Audio model.

| Decoding Strategies | POPE | Accuracy (%) | Precision (%) | Recall (%) | F1 Score (%) | Yes Rate (%) |
|---|---|---|---|---|---|---|
| Sample | Random | 27.80 | 5.17 | 6.47 | 5.75 | 36.20 |
| | Popular | 19.50 | 2.46 | 2.87 | 2.65 | 39.10 |
| | Adversarial | 17.70 | 2.79 | 3.64 | 3.16 | 32.00 |
| Greedy | Random | 29.30 | 1.92 | 2.35 | 2.11 | 37.50 |
| | Popular | 15.90 | 1.62 | 1.92 | 1.76 | 38.40 |
| | Adversarial | 15.60 | 1.08 | 1.34 | 1.20 | 35.30 |

Table 4: Results of generative tasks on instruction-tuned Qwen2-Audio model.

| Tasks | Decoding Strategies | $ECHO_I \downarrow$ (%) | $ECHO_S \downarrow$ (%) | Cov $\uparrow$ (%) |
|---|---|---|---|---|
| Audio Captioning | Sample | 84.58 | 98.26 | 12.72 |
| | Greedy | 83.91 | 100.00 | 11.66 |
| Noisy Speech Recognition | Sample | 89.28 | 91.62 | 13.81 |
| | Greedy | 83.18 | 84.46 | 12.89 |

As shown in Table 3, the model exhibits a pronounced *yes-bias*: precision remains in the low single digits and F1 stays below 6% across all negative strategies, and it affirms object presence roughly one third of the time even on adversarial negatives. Sampling improves results slightly over greedy decoding, but the bias persists. Generative results in Table 4 reveal similarly severe issues: over 80% of mentioned objects are unsupported by the audio, and nearly every output contains at least one hallucinated mention (high ECHO-I/ECHO-S), while coverage of ground-truth content remains at approximately 12%.. Sampling marginally improves captioning coverage but can worsen hallucination on noisy speech recognition, and greedy shows the opposite trade-off, indicating that decoding alone does not remedy the problem.

These results indicate that the current AUDIO-FLAN release lacks explicit hard negatives, ambiguous/null-evidence cases, and refusal-permitting prompts. As a result, models often respond confidently without sufficient acoustic support. Thus, while AUDIO-FLAN improves instruction-following ability and understanding, it does not by itself eliminate hallucination. Going forward, we will augment AUDIO-FLAN with hard negatives and uncertainty-aware/null-evidence examples, add refusal-friendly instructions with appropriate scoring, and incorporate contrastive audio–text pairs and adversarial perturbations to curb reliance on language priors.

## 4 CONCLUSION

We introduced AUDIO-FLAN, a large-scale instruction-following corpus that unifies speech, music, and general audio under a unified schema and supports both *understanding* (audio→text) and *generation* (text→audio, audio→audio, audio+text→audio). The dataset comprises 108.5M instruction–instance pairs spanning 23 major and 80 minor tasks from 52 sources. Empirically, instruction tuning on a small subset of AUDIO-FLAN yields consistent improvements on diverse understanding tasks, including zero-shot generalization to unseen tasks. The generation-side evaluation validates AUDIO-FLAN as an effective benchmark. The current release is speech-heavy, instruction-to-audio coverage outside speech is limited, and licensing constraints mean some entries are metadata-only. Hallucination persists (e.g., yes-bias and audio-unsupported mentions), and evaluation relies on an LLM-assisted normalization pipeline with targeted human checks. We will expand music and non-speech sound coverage, especially generation task, and mitigate hallucination via hard negatives, null-evidence/refusal-aware instructions, and adversarial/contrastive variants, alongside balanced curricula and uncertainty-aware training.

ETHICS STATEMENT

We curate datasets only from sources with publicly stated licenses, and we do not recruit or interact with new human subjects. Our release contains *Varied Instance Records (VIRs)* in JSONL format that are programmatic derivatives of existing public materials, including text expansions and instruction reformulations. To respect licensing terms and privacy constraints, we do not redistribute any raw audio files at this time. To reduce re-identification risk, we omit fields that directly identify individuals and use non-reversible pseudonymous identifiers where needed. We follow source de-identification practices and discourage any attempt at re-identification. We disclose limited use of large language models to generate textual variants from existing annotations and to assist with formatting. No sensitive personal data were included in prompts. We ask downstream users to respect the original licenses and applicable laws, avoid re-identification and harmful uses, and handle the materials with the same privacy and attribution safeguards described above.

REPRODUCIBILITY STATEMENT

We release the processed JSONL corpus with train, dev, test, and held-out zero-shot configurations. The data pipeline, including task taxonomy, unified instruction schema, data processing, instruction variation are documented in Section 2. Task definition and full dataset list for each minor task are described in Appendix A.1 and A.7. Training settings for instruction-tuned understanding, covering pre-trained model, LoRA configuration, optimizer/scheduler, batch sizes and learning rates, are detailed Section 3.1. Our evaluation methodology specifies the LLM-based extraction and failure handling, and the expert review protocol, metric definitions are presented in Section 3.2. Upon publication we will release the complete dataset construction pipeline and instruction-tuned checkpoints.

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

# A APPENDIX

## USE OF LARGE LANGUAGE MODELS

Large language models (LLMs) were used in three limited roles. During dataset construction, we used GPT to initialize a seed pool of record-level variants, and LLaMA to expand this pool by generating paraphrases of the records. During evaluation, we employed LLaMA in an LLM-as-judge setup to extract the expected outputs from responses produced by our instruction-tuned Qwen2-Audio model. In addition, GPT is used as a writing assistant for language editing to improve the clarity, grammar, and wording of the manuscript. All scientific ideation, experimental design, and analysis were conceived and performed exclusively by the human authors. LLMs did not determine research questions, model architectures, or conclusions.

## APPENDIX OVERVIEW

This appendix provides expanded details that complement the main paper. Appendix A.1 presents the complete task taxonomy and task definitions. Appendix A.2 illustrates examples of the Unified Task Template (UTT), while Appendix A.3 shows representative Template-Instantiated Records (TIRs). Appendix A.4 and Appendix A.5 describe the prompts used for record variation and corresponding examples of varied instances. Appendix A.6 reports detailed statistics of task instances. Finally, Appendix A.7 lists the datasets associated with each minor task together with their respective audio hours.

## A.1 TASK TAXONOMY AND DEFINITION

### Speech Domain

Here, we provide a detailed list of each minor task definition for the speech, music, and audio domains, respectively.

**Speech Recognition (3 minor tasks)**

1. Automatic Speech Recognition: transcribing speech into text.

2. Dialect Automatic Speech Recognition: Automatic Speech Recognition adapted for dialectal variations.

3. Phonetic Recognition: identifying and classifying the smallest units of sound in spoken language, known as phonemes.

**Spoken Language Understanding (2 minor tasks)**

1. Intent Classification: determining the purpose behind a user's spoken input.

2. Speech to Text Translation: translating spoken language into written text in a different language.

**Paralinguistic Attribute Recognition (7 minor tasks)**

1. Gender Recognition: classifying the biological gender of a speaker based on acoustic features of their voice. This task leverages acoustic features of speech, such as pitch, formant frequencies, and speech patterns, which tend to differ between male and female speakers due to physiological differences in the vocal tract and larynx.

2. Age Prediction: estimating the age of a speaker based on the acoustic properties of their voice. This task utilizes various speech features, such as pitch, speaking rate, formant frequencies, and spectral characteristics, which can provide cues about the speaker's age.

3. Emotion Recognition: identifying and classifying the emotional state of a speaker based on their vocal expressions.

4. Accent Recognition: identifying the regional or cultural accent of a speaker based on their speech characteristics.

5. Spoken Paragraph Recognition: determining whether two audio recordings contain the same spoken paragraph by analyzing the linguistic content.

Table 5: Task taxonomy in Audio-FLAN.

| Domain | Major Task | Minor Task |
|---|---|---|
| Speech | Speech Recognition (3) | Automatic Speech Recognition (ASR), Dialect ASR, Phonetic Recognition |
| | Spoken Language Understanding (2) | Intent Classification, Speech to Text Translation |
| | Paralinguistic Attribute Recognition (7) | Gender, Age, Emotion, Accent Recognition, Spoken Paragraph Recognition, Language ID, Dialect ID |
| | Speaker Recognition (4) | Verification, Diarization, Extraction, Identification |
| | Speech Caption (1) | Speech Caption |
| | Speech Detection (3) | Deepfake Detection, Vocoder Type Classification, Device Recognition |
| | Speech Enhancement (5) | Denoising, Dereverberation, Declipping, Bandwidth Extension, SNR Estimation |
| | Speech Generation (9) | Text-to-speech (TTS), Zero-shot TTS, Emotional TTS, Zero-shot Emotional TTS, Descriptive Speech Synthesis, Spontaneous TTS, Voice Conversion, Emotion Conversion, Speech to Speech Translation |
| **Total** | **8 major tasks** | **34 minor tasks** |
| Music | Global MIR (10) | Key Detection, Scale Recognition, Tagging, Genre Classification, Emotion Classification, Pitch Classification, Instrument Classification, Vocal Technique Classification, Artist Identification |
| | Sequential MIR (3) | Beat Tracking, Chord Estimation, Progression Extraction |
| | Single Music Reasoning (2) | Beat-level Instrument Recognition, Beat-level Pitch Estimation |
| | Multiple Music Reasoning (5) | Tempo Comparison, Instrument Comparison, Key Comparison, Technique Comparison, Emotion Comparison |
| | Music Caption (1) | Music Caption |
| | Music Separation (2) | Melody Extraction, Text-guided Source Separation |
| | Music Generation (5) | Text-to-music, Music Continuation, Lyrics-to-song, Singing Voice Synthesis, Singing Voice Conversion |
| **Total** | **7 major tasks** | **28 minor tasks** |
| Audio | Audio Event Recognition (4) | Sound Event Sequence Recognition, Event Recognition, Sound Event Detection, Acoustic Scene Classification |
| | Audio Caption (1) | Audio Caption |
| | Audio Advanced Understanding (1) | Sound Event Understanding |
| | Audio Detection (2) | Deepfake Audio Detection, Voice Activity Detection |
| | Audio Classification (2) | Speech/Silence/Music/Noise Classification, Speech/Nonspeech Detection |
| | Audio Enhancement (2) | Audio Inpainting, Audio Super-resolution |
| | Audio Separation (3) | Text-guided Source Separation, Label-querying Sound Extraction, Audio-querying Sound Extraction |
| | Audio Generation (3) | Text-guided Audio Generation, Time-grounded Text-to-Audio Generation, Audio Continuation |
| **Total** | **8 major tasks** | **18 minor tasks** |
| **Total** | **23 major tasks** | **80 minor tasks** |

6. Language Identification: determining the language spoken from a given audio sample.

7. Dialect Identification: determining the specific dialect or regional variation of a language spoken in a given audio sample.

**Speaker Recognition (4 minor tasks)**

1. Speaker Verification: verifying a speaker's identity by comparing their voice to a pre-recorded voiceprint (voice model) of the claimed identity. This process is used to authenticate or verify a speaker's identity, ensuring that the person speaking is who they claim to be. It includes text-independent and text-dependent speaker verification.

2. Speaker Diarization: identifying "who spoke when" in an audio recording containing multiple speakers. This task segments an audio stream into homogeneous regions according to the speaker identity, effectively attributing each segment of speech to its corresponding speaker.

3. Speaker Extraction: extracting the speech of a target speaker from a mixture of sounds that may include multiple speakers and background noise.

4. Speaker Identification: identifying a speaker from a set of known speakers based on their voice characteristics.

**Speech Caption (1 minor task)**

1. Speech Caption: generating synchronized text captions from spoken language.

**Speech Detection (3 minor tasks)**

1. Deepfake Detection: detecting whether an audio clip has been artificially manipulated or synthesized using AI techniques, such as voice cloning or deepfake speech generation.

2. Vocoder Type Classification: identifying and categorizing the type of vocoder used in a given speech signal.

3. Vocoder Type Classification: identifying the device used to record a given speech segment based on its acoustic features.

**Speech Enhancement (5 minor tasks)**

1. Denoising: removing unwanted noise from an audio signal to enhance the clarity and quality of the speech. This task involves distinguishing between the speech signal and the background noise, which can include sounds like traffic, machinery, conversations, or other environmental noises.

2. Dereverberation: reducing or eliminating the effects of reverberation from an audio signal. Reverberation occurs when sound waves reflect off surfaces such as walls, ceilings, and floors, causing the original speech signal to be combined with multiple delayed copies of itself.

3. Declipping: restoring audio signals that have been distorted due to clipping. Clipping occurs when the amplitude of an audio signal exceeds the maximum limit that a recording or playback system can handle, causing the peaks of the waveform to be "clipped" off.

4. Speech Bandwidth Extension: enhancing narrowband speech quality by extending its frequency range. Narrowband speech often lacks the higher frequencies that contribute to the naturalness and clarity of speech.

5. Signal-to-noise Ratio Estimation: quantifying the ratio of the power of a signal to the power of background noise. This task provides a quantitative measure of the quality of a signal.

**Speech Generation (9 minor tasks)**

1. Text to Speech: converting written text into spoken words. It involves synthesizing speech that is natural and understandable, enabling computers to "read" text aloud.

2. Zero-shot Text to Speech/Voice Cloning: generating synthetic speech for voices or styles it has never encountered during training.

3. Emotional Text to Speech: synthesizing speech with emotional nuances. The goal is to produce speech that not only conveys the content of the text but also expresses specific emotions, making the synthetic voice more engaging and human-like.

4. Zero-shot Emotional Text to Speech: generating emotional speech that adapts to an unseen speaker's voice while rendering specified emotions.

5. Descriptive Speech Synthesis: generating synthetic speech that not only replicates the spoken content but also conveys descriptive information about the context of the speech, such as emotions, tone, or other paralinguistic features.

6. Spontaneous Text to Speech: generating synthetic speech that mimics the characteristics of spontaneous unscripted human speech. Spontaneous TTS aims to replicate the naturalness, variability, and informal aspects of everyday conversational speech. This includes features such as hesitations, fillers (e.g., "um," "uh"), varying speech rates, and natural prosody changes.

7. Voice Conversion: converting one speaker's voice to resemble another's while preserving linguistic content and prosody.

8. Emotion Conversion: transforming the emotional tone of a spoken utterance from one emotion to another while preserving the linguistic content.

9. Speech to Speech Translation: converting spoken language in one language directly into spoken language in another language.

**Music Domain**

**Global MIR (10 minor tasks)**:

1. Key Detection: recognizing the key signature of the given music.

2. Scale Recognition: recognizing the scale of the given music.

3. Music Tagging: assigning descriptive tags to audio files, such as genre, style, tempo, key, artist, and emotion.

4. Genre Classification: categorizing the music into certain genres.

5. Emotion Classification: recognizing emotion categories from the music.

6. Pitch Classification: classifying the pitch of the given audio.

7. Instrument Classification: identifying all existing instruments from the music.

8. Vocal Technique Classification: detecting the playing techniques used in the vocal music.

9. Instrumental Technique Classification: detecting the playing techniques used in the instrumental music.

10 Artist Identification: identifying the relevant artists of a piece of music, given a set of artists as the options.

**Sequential MIR (3 minor tasks)**

1. Beat Tracking: detecting and aligning beats of a music excerpt.

2. Chord Estimation: estimating the chords sequence at each time step of a music excerpt.

3. Progression Extraction: extracting the chord progression represented by chord number sequence.

**Single Music Reasoning (2 minor tasks)**

1. Beat-level Instruments Recognition: recognizing the instruments from a certain beat or a certain segment.

2. Beat-level Pitch Estimation: estimating the pitch of a certain beat or segment.

**Multiple Music Reasoning (5 minor tasks)**

1. Tempo Comparison: comparing the tempo characteristics between two music excerpts.

2. Instruments Comparison: comparing instruments of two music excerpts.

3. Key Comparison: comparing keys of two music excerpts.

4. Instrumental Technique Comparison: comparing playing techniques of two music excerpts.

5. Emotion Comparison: comparing emotions of two excerpts.

**Music Caption (1 minor task)**

1. Music Caption: generating textual descriptions for a piece of music.

**Music Separation (2 minor tasks)**

1. Melody Extraction: extracting the melody at each time step from a music excerpt.

2. Text-guided Source Separation: separate certain tracks from a piece of mixed music with the text instruction.

**Music Generation (5 minor tasks)**

1. Text-to-Music Generation: generating the music given the text caption.

2. Text-guided Music Continuation: extending a given initial audio segment based on a textual description of musical characteristics while ensuring continuity and coherence.

3. Lyrics-to-song Generation: composing a song with the vocal track and instrumental track based on the given lyrics.

4. Singing Voice Synthesis: synthesizing the voice given the pitches and lyrics sequence.

5. Singing Voice Conversion: transforming the vocals (including the lyrics and melody) of singer A(source vocals) to sound like Singer B (target singer).

### Audio Domain

**Audio Event Recognition (4 minor tasks)**

1. Sound Event Sequence Recognition: identifying and sequencing various sounds in an audio stream.

2. Sound Event Recognition: detecting and identifying a particular sound in audio data.

3. Sound Event Detection: determining when a specific sound occurs within an audio clip.

4. Acoustic Scene Classification: classifying an audio clip according to the environment it represents (e.g., park, street).

**Audio Caption (1 minor task)**

1. Audio Caption: generating natural language descriptions that summarize or explain the content of an audio clip.

**Audio Advanced Understanding (1 minor task)**

1. Sound Event Understanding: extracting meaningful information from multiple audio signals (e.g. What is happening in the given audio).

**Audio Detection (2 minor tasks)**

1. Deepfake Audio Detection: identifying synthetic or manipulated audio content.

2. Voice Activity Detection: identifying segments where human speech is present in the given audio.

**Audio Classification (2 minor tasks)**

1. Speech, Silence, Music and Noise Classification: distinguishing between music, speech, and various types of noise.

2. Speech and Non-speech Detection: identifying segments which contain speech or non-speech of the given audio.

**Audio Enhancement (2 minor tasks)**

1. Audio Inpainting: filling in missing parts of an audio signal.

2. Audio Super-resolution: improving the perceptual quality of an audio signal by increasing its resolution.

**Audio Separation (3 minor tasks)**

1. Text-guided Audio Source Separation: isolating specific sound sources from an audio clip based on text input.

2. Label-querying Sound Extraction: extracting sounds belonging to a predefined category from an audio mixture, given a textual label

3. Audio-querying Sound Extraction: isolating sound sources from an audio mixture based on an example audio query.

**Audio Generation (3 minor tasks)**

1. Text-guided Audio Generation: creating audio based on a textual description.

2. Time-grounded Text-to-audio Generation: generating audio content that aligns with time-specific textual descriptions.

3. Audio Continuation: extending an audio clip by generating additional content that seamlessly continues the original.

## A.2 UNIFIED TASK TEMPLATE

The Unified Task Template (UTT) is illustrated as follows:

```
Unified Task Template (UTT)

{
  "instruction": "{natural_language_instruction}",
  "input": "{text_input}\n<|SOA|>{audio_id}<|EOA|>",
  "output": "{text_label|free_text|<|SOA|>{generated_audio_id
      }<|EOA|>}",
  "uuid": "{uuid}",
  "split": "{train|dev|test}",
  "task_type": {
    "major": ["{major_task}"],
    "minor": ["{minor_task}"],
    "U/G": "{understanding|generation}",
    "unseen": "{true|false}"
  },
  "domain": "{speech|music|audio}",
  "source": ["{source_domain}"],
  "other": "{optional_metadata_or_null}"
}
```

The definitions of each field are described as follows:

**Instruction**: this field provides the instructions for the task, outlining the specific operation to be performed.

**Input**: this field contains the input data for the task, which represents the raw information to be processed.

**Output**: this field represents the expected result or outcome after processing the input data.

**Uuid**: this field assigns a unique identifier to each task instance, enabling the system to track and manage individual tasks.

**Split**: this field specifies the dataset partition for the task, such as "train", "test", or "dev", which correspond to the training, testing, and development datasets, respectively.

**Task_type**: this field outlines the nature of the task:

- **Major**: indicates the primary category of the task.

- **Minor**: specifies the secondary or more specific task.

- **U/G**: distinguishes whether the task focuses on generation or understanding.

- **Unseen**: a boolean value that indicates whether the task involves data that has not been encountered before.

**Domain**: this field defines the domain in which the task is situated, such as "speech", "music", or "audio".

**Source**: this field identifies the origin of the audio, such as "audiobook", "youtube", or "studio", signifying where the audio signal is sourced from.

**Other**: this field can store any additional metadata relevant to the task, if applicable.

## A.3 Template-instantiated records

Here we provide three Template -Instantiated Records (TIRs) for three different tasks in JSONL format with all fields.

**Speech-to-Text Translation**

```
{"instruction": "Please translate the speech into the text in
    English.",
  "input": "<|SOA|>Speech_Audio<|EOA|>",
  "output": "Nevertheless, there are many distinctive ways of
      drinking coffee around the world that are worth
      experiencing.",
  "uuid": "UUID",
  "split": ["train"],
  "task_type": {
    "major": ["Spoken Language Understanding"],
    "minor": ["Speech-to-text Translation"],
    "U/G": ["understanding"],
    "unseen": false
  },
  "domain": "speech",
  "source": ["unknown"]
"other": null}
```

**Text-guided Music Continuation**

```
{"instruction": "Please continue the audio music prompt based
    on the given text description",
"input": "This is a Carnatic music piece set in the atana
    raga. It follows the 5/8 meter and is composed in the
    khandaChapu taala. The lead instrument featured in this
    performance is vocal, accompanied by Mridangam. The kalai
    of this composition is 1.\n audio prompt: <|SOA|>
    Music_Audio<|EOA|>",
"output": "audio: <|SOA|>Musi_Audio<|EOA|>",
"uuid": "UUID",
"split": ["test"],
"task_type": {
    "major": ["Music Generation"],
    "minor": ["Text-guided Music Continuation"],
    "U/G": ["generation"],
    "unseen": false
    },
"domain": "music",
"source": ["unknown"],
"other": null}
```

**Sound Super-resolution**

```
{"instruction": "Please increase the resolution of the given
    audio signal to 32k Hz.",
"input": "audio: <|SOA|>Sound_Audio<|EOA|>.",
"output": "<|SOA|>Sound_Audio<|EOA|>",
```

```
"uuid": "UUID",
"split": ["train"],
"task_type": {
    "major": ["Sound Generation"],
    "minor": ["Sound Super-resolution"],
    "U/G": ["generation"],
    "unseen": false
    },
"domain": "audio",
"source": ["youtube"],
"other": null}
```

## A.4  RECORD VARIATION PROMPT

As mentioned in Section 2.4, all Template-Instantiated Records (TIRs) are varied by LLAMA-3.1-70B-INSTRUCT with the variation prompt as follows.

---

You are tasked with paraphrasing the values of the following fields: "instruction", "input", and "output". Your goal is to generate varied and creative rewrites for each of these fields. Please adhere to the following guidelines:

1. **Paraphrase Instructions**:

   - Paraphrase the "instruction" field in diverse ways by changing the sentence structure, style, and tone. Use a variety of sentence types, including:
     - Direct commands (e.g., "Turn this into speech.")
     - Polite requests (e.g., "Could you please convert this to speech?")
     - Questions (e.g., "Can you turn this into audio?")
     - Suggestions (e.g., "It would be great if you could convert this.")
     - Exclamations or emphatic forms (e.g., "I really need this to be in audio form.")
   - Feel free to add polite elements, such as "please," "kindly," or "if you would be so kind," as long as they remain natural.

2. **Paraphrase Inputs**:

   - Change the labels for fields like "text:"`, "text_description:"`, "audio:"`, "speaker_audio:"`, "audio_sample1"`, "audio_sample2"` etc., according to "instruction", while retaining their original meaning. Examples include:
     - "text:" to "spoken text," "speech input," "text excerpt," etc.
     - "text_description:" to "voice style," "descriptive text," "tone characteristics," etc.
     - "audio:" to "source audio," "reference speech," "given recording," etc.
     - "speaker_audio:" to "speaker prompt," "reference voice," "voice sample," etc.
   - Ensure that the content following "text:" remains semantically identical to the original. The content following each label should remain unchanged, with only the labels varying.

3. **Maintain Consistency in Outputs**:

   - Depending on the tone of the instruction, introduce additional phrases such as:
     - "The gender is ", "Gender: ".
     - "The language is ", "Language in the given speech is ".
     - "The speakers in the given two speechs are ", "The anwser is ".
     - "Transcription is: ", "The text of the given speech is: ".
     - "IPA Phonemes is: ", "phonemes of the given speech is: ".

---

- "Descriptive text of the given speech is: ", "The speaking style is: ", "Speech caption is: ".
  - Ensure the "output" field contains the substring `|SOA|>audio<|EOA|` and the content that follows it, preserving both the structure and meaning.
  - You may optionally introduce phrases before `|SOA|>audio<|EOA|` (e.g., "Generated speech is:", "Audio output:", "The resulting audio is:"). Avoid altering or introducing inconsistencies in the audio filename (e.g., `|SOA|>13_LibriTTS-R_260_123288_000009_000000<|EOA|`).

4. **Ensure JSON Validity**:
  - All strings must be enclosed in double quotes.
  - Key-value pairs must be separated by commas.
  - The JSON structure for "instruction", "input", and "output" fields must be valid and consistent.

5. **The number of objects in the output should match the number of inputs.** Ensure the format is valid JSON and all JSON objects are properly enclosed. Each entry should be separated by a newline and all JSON should be enclosed in an array.

Please provide the output in valid JSON format (an array of JSON objects), and ensure proper formatting.

## A.5 VARIED INSTANCE RECORDS

After the Template -Instantiated Records (shown in Appendix A.3) are diversified by the prompt in Appendix A.4, we obtain Varied Instance Records (VIRs), which are shown below.

---

**Speech-to-Text Translation (VIR Example 1)**

```
{
"instruction": "Would you be able to convert the spoken words
    into English text?",
"input": "<|SOA|>Speech_Audio<|EOA|>",
"output": "Nevertheless, there are many distinctive ways of
    drinking coffee around the world that are worth
    experiencing.",
}
```

---

**Speech-to-Text Translation (VIR Example 2)**

```
{
"instruction": Please provide the English translation of the
    audio speech.",
"input": "<|SOA|>Speech_Audio<|EOA|>",
"output": "Nevertheless, there are many distinctive ways of
    drinking coffee around the world that are worth
    experiencing.",
}
```

```
Speech-to-Text Translation (VIR Example 3)

{
"instruction": Could you kindly translate the given speech
    into written English?",
"input": "<|SOA|>Speech_Audio<|EOA|>",
"output": "Nevertheless, there are many distinctive ways of
    drinking coffee around the world that are worth
    experiencing.",
}
```

## A.6 DETAILED INSTANCES STATISTICS

Detailed instance statistics of each major task in speech, music, and audio domains are presented.

Table 6: Detailed statistics of tasks and instances in Audio-FLAN. "U/G" denotes whether the task is for understanding (U) or generation (G). Tasks with audio outputs are marked as generation.

| Domain | Major Task | # Minor Tasks | # Instances | Input/Output | U/G |
|---|---|---|---|---|---|
| Speech | Speech Recognition | 3 | 12.05M | audio/text | U |
| | Spoken Language Understanding | 2 | 26.25M | audio/text | U |
| | Paralinguistic Attribute Recognition | 7 | 16.47M | audio/text | U |
| | Speaker Recognition | 4 | 0.73M | audio/text | U |
| | Speech Caption | 1 | 0.35M | audio/text | U |
| | Speech Detection | 3 | 1.57M | audio/text | U |
| | Speech Enhancement | 5 | 1.48M | audio/audio | G |
| | Speech Generation | 9 | 41.52M | (audio, text)/audio | G |
| **Total** | 8 | **34** | **100.42M** | - | - |
| Music | Global MIR | 10 | 0.34M | audio/text | U |
| | Sequential MIR | 3 | 0.43M | audio/text | U |
| | Single Music Reasoning | 2 | 95.86K | audio/text | U |
| | Multiple Music Reasoning | 5 | 0.57M | audio/text | U |
| | Music Caption | 1 | 28.21K | audio/text | U |
| | Music Separation | 2 | 40.26K | audio/audio | G |
| | Music Generation | 5 | 0.67M | (audio, text)/audio | G |
| **Total** | 7 | **28** | **2.17M** | - | - |
| Audio | Audio Event Recognition | 4 | 1.30M | audio/text | U |
| | Audio Caption | 1 | 0.82M | audio/text | U |
| | Audio Advanced Understanding | 1 | 10K | audio/text | U |
| | Audio Detection | 2 | 1.08M | audio/text | U |
| | Audio Classification | 2 | 0.38M | audio/text | U |
| | Audio Enhancement | 2 | 0.15M | audio/audio | G |
| | Audio Separation | 3 | 0.89M | audio/audio | G |
| | Audio Generation | 3 | 1.31M | (audio, text)/audio | G |
| **Total** | 8 | **18** | **5.91M** | - | - |
| **Total** | **23** | **80** | **108.50M** | - | - |

## A.7 DATASETS INFORMATION FOR EACH TASK

Here, we present the dataset sources used for each minor task in Table 7 and the hours of each dataset 8.

Table 7: Minor task and its corresponding datasets.

| Domain | Minor Task | Dataset |
|--------|-----------|---------|
| Speech | Automatic Speech Recognition | Aishell1 (Bu et al., 2017), Aishell2 (Du et al., 2018), Aishell3 (Shi et al., 2020), ESD (Zhou et al., 2022), EmoV_DB (Adigwe et al., 2018), FLEURS (Conneau et al., 2023), Fluent Speech Commands (Lugosch et al., 2019), HQ-Conversations (Xia et al., 2024), HiFi TTS (Bakhturina et al., 2021), LJSpeech (Ito & Johnson, 2017), MLS (Pratap et al., 2020), The Parallel Audiobook Corpus (Ribeiro, 2018), VCTK (Veaux et al., 2017), aidatatang (Beijing DataTang Technology Co., n.d.), common voice (Ardila et al., 2019), LibriTTS-R (Koizumi et al., 2023) |
| | Dialect Automatic Speech Recognition | KeSpeech (Tang et al., 2021) |
| | Phonetic Recognition | Aishell3 (Shi et al., 2020), LibriTTS-R (Koizumi et al., 2023) |
| | Intent Classification | Fluent Speech Commands (Qian et al., 2021) |
| | Gender Recognition | Aishell1 (Bu et al., 2017) (Bu et al., 2017), Aishell2 (Du et al., 2018), Aishell3 (Shi et al., 2020), Fluent Speech Commands (Lugosch et al., 2019), HQ-Conversations (Xia et al., 2024), KeSpeech (Tang et al., 2021), The Parallel Audiobook Corpus (Ribeiro, 2018), LibriTTS-R (Koizumi et al., 2023) |
| | Age Recognition | HQ-Conversations (Xia et al., 2024), KeSpeech (Tang et al., 2021) |
| | Emotion Recognition | ESD (Zhou et al., 2022) |
| | Accent Recognition | HQ-Conversations (Xia et al., 2024) |
| | Spoken Paragraph Recognition | LibriTTS-R (Koizumi et al., 2023) |
| | Language Identification | Aishell1 (Bu et al., 2017) (Bu et al., 2017), Aishell2 (Du et al., 2018), Aishell3 (Shi et al., 2020), ESD (Zhou et al., 2022), EmoV_DB (Adigwe et al., 2018), FLEURS (Conneau et al., 2023), HQ-Conversations (Xia et al., 2024), HiFi TTS (Bakhturina et al., 2021), LJSpeech (Ito & Johnson, 2017), MLS (Pratap et al., 2020), The Parallel Audiobook Corpus (Ribeiro, 2018), aidatatang (Beijing DataTang Technology Co., n.d.), common voice (Ardila et al., 2019), LibriTTS-R (Koizumi et al., 2023) |
| | Dialect Identification | KeSpeech (Tang et al., 2021) |
| | Speaker Verification | Aishell1 (Bu et al., 2017) (Bu et al., 2017), Aishell2 (Du et al., 2018), Aishell3 (Shi et al., 2020), ESD (Zhou et al., 2022), EmoV_DB (Adigwe et al., 2018), Fluent Speech Commands (Lugosch et al., 2019), HQ-Conversations (Xia et al., 2024), HiFi TTS (Bakhturina et al., 2021), KeSpeech (Tang et al., 2021), The Parallel Audiobook Corpus (Ribeiro, 2018), LibriTTS-R (Koizumi et al., 2023) |

| Domain | Minor Task | Dataset |
|---|---|---|
| | *Continued from the previous page* | |
| | Speaker Diarization | AliMeeting (Yu et al., 2022) |
| | Speaker Extraction | LibriMix (Cosentino et al., 2020) |
| | Speaker Identification | KeSpeech (Tang et al., 2021) |
| | Speech Caption | LibriTTS-R (Koizumi et al., 2023) |
| | Deepfake Detection | ASVSpoof2021 (Liu et al., 2023) |
| | Vocoder Type Classification | ASVSpoof2021 (Liu et al., 2023) |
| | Device Recognition | HQ-Conversations (Xia et al., 2024) |
| | Denoising | DNS (Reddy et al.) |
| | Dereverberation | DNS (Reddy et al.) |
| | Declipping | DNS (Reddy et al.) |
| | Speech Bandwidth Extension | DNS (Reddy et al.) |
| | Signal-to-noise Ratio Estimation | LibriTTS-R (Koizumi et al., 2023) |
| | Speech to Text Translation | CVSS (Jia et al., 2022), FLEURS (Conneau et al., 2023) |
| | Text to Speech | Aishell1 (Bu et al., 2017) (Bu et al., 2017), Aishell2 (Du et al., 2018), Aishell3 (Shi et al., 2020), ESD (Zhou et al., 2022), EmoV_DB (Adigwe et al., 2018), FLEURS (Conneau et al., 2023), Fluent Speech Commands (Lugosch et al., 2019), HQ-Conversations (Xia et al., 2024), HiFi TTS (Bakhturina et al., 2021), KeSpeech (Tang et al., 2021), LJSpeech (Ito & Johnson, 2017), MLS (Pratap et al., 2020), The Parallel Audiobook Corpus (Ribeiro, 2018), VCTK (Veaux et al., 2017), aidatatang (Beijing DataTang Technology Co., n.d.), common voice (Ardila et al., 2019), LibriTTS-R (Koizumi et al., 2023), Genshin (AI-Hobbyist, 2024a), StarRail (AI-Hobbyist, 2024b) |
| | Zero-shot Text to Speech | Fluent Speech Commands (Lugosch et al., 2019), LibriTTS-R (Koizumi et al., 2023),HQ-Conversations (Xia et al., 2024),Fluent Speech Commands (Lugosch et al., 2019), Aishell2 (Du et al., 2018), Aishell3 (Shi et al., 2020), KeSpeech (Tang et al., 2021) |
| | Emotional Text to Speech | ESD (Zhou et al., 2022), EmoV_DB (Adigwe et al., 2018) |
| | Zero-shot Emotional Text to Speech | ESD (Zhou et al., 2022) |
| | Descriptive Speech Synthesis | LibriTTS-R (Koizumi et al., 2023) |
| | Voice Conversion | ESD (Zhou et al., 2022) |
| | Emotion Conversion | ESD (Zhou et al., 2022) |
| | Speech to Speech Translation | FLEURS (Conneau et al., 2023) |
| Music | Key Detection | AAM (Ostermann et al., 2023), FreeSound Loop Dataset (Ramires et al., 2020) |
| | Music Tagging | MTG (Bogdanov et al., 2019) |
| | Genre Classification | CSD (Choi et al., 2020), MTG (Bogdanov et al., 2019),FreeSound Loop Dataset (Ramires et al., 2020) |
| | Emotion Classification | MTG (Bogdanov et al., 2019) |
| | Pitch Classification | NSynth (Engel et al., 2017) |
| | Instrument Classification | AAM (Ostermann et al., 2023), MTG (Bogdanov et al., 2019), NSynth (Engel et al., 2017) |

| Domain | Minor Task | Dataset |
|---|---|---|
| | | *Continued from the previous page* |
| | Vocal Technique Classification | VocalSet (Wilkins et al., 2018) |
| | Instrumental Technique Classification | CCOM-HuQin (Zhang et al., 2022b) |
| | Artist Identification | FMA (Defferrard et al., 2016) |
| | Beat Tracking | AAM (Ostermann et al., 2023) |
| | Melody Extraction | MedleyDB (Bittner et al., 2014) |
| | Chord Estimation | AAM (Ostermann et al., 2023) |
| | Beat-level Instrument Recognition | AAM (Ostermann et al., 2023) |
| | Progression Extraction | JazzNet (Adegbija, 2023) |
| | Scale Recognition | JazzNet (Adegbija, 2023) |
| | Beat-level Pitch Estimation | AAM (Ostermann et al., 2023), CSD (Choi et al., 2020), Vocadito (Bittner et al., 2021) |
| | Tempo Comparison | GTZAN Rhythm (Marchand et al., 2015), FreeSound Loop Dataset (Ramires et al., 2020) |
| | Instrument Comparison | NSynth (Engel et al., 2017) |
| | Key Comparison | GiantSteps Key (Knees et al., 2015) |
| | Emotion Comparison | MTG (Bogdanov et al., 2019) |
| | Instrumental Technique Comparison | CCOM-HuQin (Zhang et al., 2022b) |
| | Music Caption | Musiccaps (Agostinelli et al., 2023), FreeSound Loop Dataset (Ramires et al., 2020) |
| | Text-to-music Generation | FreeSound Loop Dataset (Ramires et al., 2020), Musiccaps (Agostinelli et al., 2023), Compmusic (Srinivasamurthy et al., 2021; Anantapadmanabhan et al., 2013; Black et al., 2014; Caro Repetto, 2018; Gupta et al., 2015; Koduri et al., 2014; Kuriakose et al., 2015; Pretto et al., 2018; Srinivasamurthy & Serra, 2014; Srinivasamurthy et al., 2015; 2016) |
| | Text-guided Music Continuation | Compmusic (Srinivasamurthy et al., 2021; Anantapadmanabhan et al., 2013; Black et al., 2014; Caro Repetto, 2018; Gupta et al., 2015; Koduri et al., 2014; Kuriakose et al., 2015; Pretto et al., 2018; Srinivasamurthy & Serra, 2014; Srinivasamurthy et al., 2015; 2016) |
| | Lyrics2song Generation | CSD (Choi et al., 2020), Vocadito (Bittner et al., 2021), Opencpop (Wang et al., 2022), Opensinger (Huang et al., 2021) |
| | Singing Voice Synthesis | CSD (Choi et al., 2020), Vocadito (Bittner et al., 2021), Opencpop (Wang et al., 2022), Opensinger (Huang et al., 2021) |
| | Singing Voice Conversion | Opensinger (Huang et al., 2021), m4singer (Zhang et al., 2022a) |
| | Text-guided Source Separation | MedleyVox (Jeon et al., 2023), Moises (Pereira et al., 2023) |
| Audio | Sound Event Sequence Recognition | Audioset (Gemmeke et al., 2017) |
| | Acoustic Scene Classification | TAU Urban Acoustic Scenes (Heittola et al., 2020) |
| | Audio Caption | Audioset (Gemmeke et al., 2017), Freesound (Font et al., 2013) |
| | Text-guided Audio Generation | Audioset (Gemmeke et al., 2017), Freesound (Font et al., 2013) |
| | Time-grounded Text-to-audio Generation | Audioset (Gemmeke et al., 2017) |
| | Audio Continuation | Wavcaps (Mei et al., 2024) |

| Domain | Minor Task | Dataset |
|---|---|---|
| | *Continued from the previous page* | |
| | Audio Inpainting | Audioset (Gemmeke et al., 2017) |
| | Audio Super-resolution | Audioset (Gemmeke et al., 2017) |
| | Sound Event Understanding | Vocal Imitation (Kim et al., 2018) |
| | Text-guided Audio Source Separation | Wavcaps (Mei et al., 2024) |
| | Label-querying Sound Extraction | VGG (Chen et al., 2020) |
| | Audio-querying Sound Extraction | VGG (Chen et al., 2020) |
| | Deepfake Audio Detection | ADD2023 (Yi et al., 2023) |
| | Voice Activity Detection | DNS for VAD (Reddy et al.) |
| | Speech, Silence, Music and Noise Classification | Audioset (Gemmeke et al., 2017) |
| | Speech Nonspeech Detection | Wavcaps (Mei et al., 2024) |

Table 8: Detailed information of datasets.

| Domain | Dataset | Audio Length (#hours) |
|---|---|---|
| Speech | common voice (Ardila et al., 2019) | 19,673 |
| | aidatatang (Beijing DataTang Technology Co., n.d.) | 200 |
| | libritts-R (Koizumi et al., 2023) | 585 |
| | libritts (Zen et al., 2019) | 586 |
| | HQ-Conversations (Xia et al., 2024) | 100 |
| | EmoV_DB (Adigwe et al., 2018) | 9.49 |
| | VCTK (Veaux et al., 2017) | 44 |
| | MLS (Pratap et al., 2020) | 45,042 |
| | FLEURS (Conneau et al., 2023) | 17 |
| | Fluent speech commands (Lugosch et al., 2019) | 19 |
| | LibriMix (Cosentino et al., 2020) | 500 |
| | Aishell1 (Bu et al., 2017) | 155 |
| | Aishell2 (Du et al., 2018) | 1,036 |
| | Aishell3 (Shi et al., 2020) | 65 |
| | LJSpeech (Ito & Johnson, 2017) | 23.9 |
| | The Parallel Audiobook Corpus (Ribeiro, 2018) | 121 |
| | HiFi TTS (Bakhturina et al., 2021) | 291.6 |
| | KeSpeech (Tang et al., 2021) | 1,428 |
| | ESD (Zhou et al., 2022) | 29 |
| | CVSS (Jia et al., 2022) | 3,809 |
| | ASVSpoof2021 (Liu et al., 2023) | 1270.5 |
| Music | Opencpop (Wang et al., 2022) | 5.2 |
| | m4singer (Zhang et al., 2022a) | 29.77 |
| | FreeSound Loop Dataset (Ramires et al., 2020) | 34.7 |
| | Opensinger (Huang et al., 2021) | 50 |
| | MedleyVox (Jeon et al., 2023) | 1.1 |
| | Vocadito (Bittner et al., 2021) | 0.23 |
| | MoisesDB (Pereira et al., 2023) | 14.4 |
| | CSD (Choi et al., 2020) | 4.86 |
| | Musiccaps (Agostinelli et al., 2023) | 15.28 |
| | GTZAN rhythm (Marchand et al., 2015) | 8.3 |
| | GiantSteps key (Knees et al., 2015) | 20.07 |
| | CCOM-HuQin (Zhang et al., 2022b) | 4.3 |
| | NSynth (Engel et al., 2017) | 340 |
| | MedleyDB (Bittner et al., 2014) | 7.45 |
| | Free Music Archive (Defferrard et al., 2017) | 8,232 |
| | AAM (Ostermann et al., 2023) | 125 |
| | MTG (Bogdanov et al., 2019) | 3,777 |
| Audio | Audioset (Gemmeke et al., 2017) | 5208 |
| | VGGSound (Chen et al., 2020) | 550 |
| | Wavcaps (Mei et al., 2024) | 3,793.3 |
| | freesound (Font et al., 2013) | 6446.05 |
| | TAU Urban Acoustic Scenes (Heittola et al., 2020) | 68.18 |
| | Vocal Imitation (Kim et al., 2018) | 24 |
| | ESC (Piczak) | 2.78 |
| | DNS for VAD (Reddy et al.) | 562.72 |
| | ADD2023 (Yi et al., 2023) | 220 |

