# OpenReview forum: "Audio-FLAN: An Instruction-Following Dataset for Unified Understanding and Generation of Speech, Music, and Sound"
_ICLR.cc/2026/Conference — Submitted to ICLR 2026_

### Official Review · Reviewer_eRX3 · 2025-10-23

**Soundness:** 3
**Presentation:** 4
**Contribution:** 3
**Rating:** 4
**Confidence:** 3

**Summary:**

This paper released a large (108.5M) speech instruction following dataset, including both speech generation and understanding. Each example follows the schema of (instruction, input, output). Specifically, there are a total of 23 major and 80 minor tasks, which are drawn from 52 datasets. Their experiments on Qwen2-Audio-7b-Instruct further show the model’s improvement on unseen tasks after tuning on a subset of their dataset, indicating the usefulness and validity of their work.

**Strengths:**

1. This paper provided a large and diverse speech instruction tuning dataset, which makes a large contribution to the field. To my knowledge, there is no dataset with this scale in the speech domain that exists before.
2. The authors conducted detailed experiments on their dataset, including zero-shot performance on unseen tasks and hallucination analysis.

**Weaknesses:**

1. If I understand it correctly, the so-called “unseen” tasks are collected using the same pipeline, same LLM. I’m not fully convinced by it as totally “unseen” or out of distribution.
2. The paper would be more robust if the author showed the improvement on other speech benchmarks after tuning on the proposed dataset.
3. It is frustrating that there are no results for audio and music generation. The author does mention that their fin-tuned model fails on these tasks. However, this makes me wonder whether there are problems with the curated data for music and audio generation tasks in this proposed dataset.
4. Lacking a topline(e.g., GPT-4/Gemini…) for all tasks. This makes it hard for the reader to judge the quality of the curated data.
5. Lacking a random baseline (probably without giving the SpeechLLM audio/speech/music input) in Table 1, which makes people question whether the model really learns non-trivial knowledge after fin-tuning on seen tasks. Especially for some tasks, the performance is quite low after tuning on seen tasks.

**Questions:**

1. Why is the author using HuBERT-Large for evaluating the WER in generation tasks. Why not Whisper models? Whisper models should be more robust in this case.
2. It would be better to add an explanation for “/” in the caption of Table1.

---

### Official Review · Reviewer_dKuG · 2025-10-25

**Soundness:** 1
**Presentation:** 3
**Contribution:** 2
**Rating:** 2
**Confidence:** 5

**Summary:**

The paper introduces AUDIO-FLAN, a large-scale instruction-following dataset that unifies understanding and generation across speech, music, and general audio under a single schema (instruction, input, output). It reports 108.5M instances spanning 23 major and 80 minor tasks aggregated from 52 sources, and provides train/dev/test splits plus zero-shot (unseen task) settings. The authors show performance gains after finetuning Qwen2-Audio.

**Strengths:**

1. The paper proposes a large-scale, unified instruction-following dataset for audio, speech, and music understanding and generation. This lays a strong foundation for training large audio-language models.

2. The paper is easy to follow.

**Weaknesses:**

Despite recognizing the motivation and contribution of this paper, the designed and conducted experiments fail to demonstrate the quality of the proposed dataset.

1. The paper claims it is the first benchmark to integrate understanding and generation. However, the evaluation method described cannot evaluate all the tasks provided. Compared to existing benchmarks like Dynamic SUPERB Phase 2, the novelty lies mainly in the generation aspect. However, many proposed tasks—such as Emotional TTS, Descriptive Speech Synthesis, Speech-to-Speech Translation, Emotion Conversion, Text-to-Music Generation, and Text-to-Audio Generation—lack automatic evaluation methods. This also raises the question of how the proposed dataset helps models improve on these tasks.
2. Why was the zero-shot evaluation only conducted on UniAudio? Experiments on SOTA models like Qwen-2.5-omni, Qwen3-omni, GPT-4o, Gemini, Mimo-Audio, or Kimi-audio would help readers realize the shortages of current State-of-the-Art (SOTA) models.
3. The paper experiments on outdated models like Qwen2-Audio. I’m wondering if the dataset can provide improvements on other SOTA models like Qwen-2.5-Omni? Please adopt Qwen-2.5-Omni or other SOTA models. it would greatly strengthen this paper.
4. The experiments are conducted on a 10% training subset. There are no experiments on the full training set to validate the effect of full data, and no ablation on models trained on different subset sizes.
5. Discussions about dataset licenses are lacking.
6. The conclusion mentions current limitations (e.g., the dataset mainly focuses on speech-related tasks), how would the authors tackle this issue?
7. Section 3.6 indicates current issues with the model release. Why didn't the authors take action to solve these problems, such as yes-bias?

**Questions:**

### Typos
1. If prior work has been accepted (e.g., AISHELL-3 at Interspeech 2021; Dynamic SUPERB Phase 2; MMAU at ICLR 2025), please cite the conference versions rather than preprints.

### Questions
1.  Is the human evaluation during dataset construction is sample-based rather than inspecting all instances? If so, please report an analysis of the human evaluation, such as the percentage of failure cases.

**Details Of Ethics Concerns:**

Discussions about dataset licenses are lacking. (Conclusion says there are still some constraints).

---

### Official Review · Reviewer_D7te · 2025-10-31

**Soundness:** 1
**Presentation:** 3
**Contribution:** 2
**Rating:** 2
**Confidence:** 4

**Summary:**

This paper present Audio-FLAN, a large-scale instruction-following dataset unifying speech, music and general audio tasks under a single schema supporting both understanding (audio->text) and generation (text/audio->audio). The dataset comprises over 100M audio-instruction pairs spanning 80 tasks. Empirical validation demonstrates that instruction-tuning on Audio-FLAN is able to improve performance on understanding and generation, respectively.

**Strengths:**

* This work contributes to the community by creating the largest unified instruction-following dataset to date, comprising 108.5M samples spanning speech, music, and audio domains. The dataset covers 80 diverse tasks, including both well-studied tasks and underexplored areas such as beat-level music reasoning, thereby expanding the scope of research in this field.
* While not originally proposed, the Self-Instruction [1] pipeline enables a semi-automated approach to scale up and diversify instruction collection while maintaining semantic consistency. This work represents the first application of this pipeline for generating audio instructions.

[1] Wang, Yizhong, et al. "Self-Instruct: Aligning Language Models with Self-Generated Instructions." ACL 2023, 2023.

**Weaknesses:**

* The purpose of creating a unified dataset is unclear if no model is jointly trained on both understanding and generation tasks. It would be valuable to see how these two training paradigms interact and affect each other’s performance.
* The experimental setup demonstrates limited evidence for the effectiveness of generation tasks, as there is no baseline comparison and the results for text-to-audio or text-to-music generation are missing.
* The evaluation design for understanding tasks is questionable, where the model is trained on only 10% of Audio-FLAN while designating certain tasks as “unseen” seems arbitrary and may not meaningfully test generalization.
* The paper does not provide any comparison with existing audio-instruction datasets [2,3] (as well as comparison with existing benchmarks), making it difficult to assess the incremental value of Audio-FLAN relative to prior efforts.


[2] Lu, Ke-Han, et al. "DeSTA2. 5-Audio: Toward General-Purpose Large Audio Language Model with Self-Generated Cross-Modal Alignment." 2025.

[3] Goel, Arushi, et al. "Audio flamingo 3: Advancing audio intelligence with fully open large audio language models." 2025.

**Questions:**

* Are there any preliminary experimental results demonstrating the effectiveness of the domain imbalance mitigation strategies mentioned in Section 2.6? What specific approach was actually applied in the current experiments?
* The Self-Instruct pipeline was originally proposed to enhance the diversity of generated instructions. How diverse are the instructions produced by the current pipeline? Has any text-based analysis (such as lexical or semantic diversity) been conducted to quantify this?

---

### Official Review · Reviewer_BHhw · 2025-11-06

**Soundness:** 3
**Presentation:** 3
**Contribution:** 2
**Rating:** 6
**Confidence:** 5

**Summary:**

This paper releases Audio-FLAN, an instruction-following corpus that unifies speech, music, and general audio for both understanding (audio-->text) and generation (text/audio-->audio). Each example follows a Unified Task Template with {instruction, input, output}; audio spans are delimited by <|SOA|>…<|EOA|>. The authors claim 108.5M instances across 23 major / 80 minor tasks from 52 sources, and provide train/dev/test plus zero-shot (unseen-task) splits. They build Self-Instruct paraphrases (GPT-4o --> seed pool; Llama-3.1-70B --> constrained rewrites) and validate automatically + spot-check manually. Experiments: (i) instruction-tune Qwen2-Audio-7B-Instruct on 10% of data (seen-task only), reporting big gains; (ii) evaluate UniAudio for generation tasks; (iii) zero-shot tests; (iv) a hallucination probe showing severe yes-bias and >80% unsupported mentions. Ethics: no raw audio is redistributed.

**Strengths:**

- First benchmark with training data for unified audio models in my understanding.
- Unified, cross-domain schema that spans speech/music/sound and both directions (U<-->G) with a single JSONL template and audio markers; this lowers friction for multi-task training.
- A large task space (23/80) intended to exercise time-sequential reasoning (e.g., beat-level MIR).
- 108.5M instruction instances spanning from 52 sources is a very novel contribution for the community.
- Although that speech dominates (100.42M speech vs 2.17M music / 5.91M general-audio) in the dataset, but the domain tagging plus per-domain stats make it feasible to rebalance sampling downstream.
- For messy, instruction-following responses, the LLM-based extractor + expert review provides a workable normalization method; they also log extraction failures for transparency.

**Weaknesses:**

- Some human verification of the generated data using Llama would definitely strengthen the paper.
- The authors use LLM as a judge for evaluating open ended generations -  a human - llm correlation is missing.
- Constrained Llama rewrites “must not alter labels/spans/timestamps,” but only spot-checks are reported.

**Questions:**

NA

---

### Official Review · Reviewer_kEqo · 2025-11-06

**Soundness:** 3
**Presentation:** 3
**Contribution:** 2
**Rating:** 4
**Confidence:** 5

**Summary:**

This paper introduces AUDIO-FLAN, a massive 108.5M-sample instruction-following dataset that unifies speech, music, and general audio under the same instruction template. Unlike most existing corpora that are either (i) speech-only or (ii) understanding-only, AUDIO-FLAN supports both understanding (audio→text) and generative tasks (text/audio/{audio,text}→audio) using a single JSON schema. It covers 23 major and 80 minor tasks collected from 52 datasets, and uses a Self-Instruct-style pipeline to produce stylistically diverse paraphrased instances while preserving semantic correctness.

The authors show (on Qwen2-Audio) that instruction tuning with just a 10% subset of AUDIO-FLAN yields large gains across speech, music, and general audio — including previously unparseable tasks becoming solvable — and that these gains transfer to zero-shot unseen tasks (especially MIR tasks). They also evaluate UNIAUDIO generation under a unified protocol, showing AUDIO-FLAN can act as a benchmark for instruction-conditioned audio generation. A hallucination study demonstrates that despite improved instruction adherence, hallucination remains severe, motivating augmentation with hard negatives.

**Strengths:**

- first OPEN dataset to unify all three audio domains + both directions (understanding + generation) under one schema
- extremely large, broad coverage; well-engineered templating and variation
- strong empirical evidence that small FT on it boosts instruction understanding and zero-shot transfer (more on this below)
- benchmark design is compatible with preexisting models without architectural changes

**Weaknesses:**

- extremely speech-heavy distribution (≈ 100M/108M) — imbalance acknowledged
- relies on LLM-assisted normalization + human adjudication in eval (adds subjectivity and cost)
- Related to the above, I do not see how hallucinations in generations were handled or reduced. or negative generations were handled
- generation evaluation only meaningfully covers speech — music generation eval lacking
- additionally only one model has been used to eval
- I do not see any insights on the actual data, like difficulty level etc. This makes it hard to determine how useful the data is for frontier model training.
- hallucination still extreme — dataset alone does not fix it
- I am not fond of the eval setup, while I acknowledge that fine-tuning from scratch is hard, but fine-tuning an open-weights model (where training data is not known) says little about the efficacy of the data. The additional problem is LoRA fine-tuning. LoRA fine-tuning does not give a model new capabilities (mentioned in several papers) -- so how is FT on the model actually helping on downstream tasks?
- I dont se e

**Questions:**

- What is the philosophy of the eval setup? how was it determined?

---

### Meta-Review · Area_Chair_ATFu · 2026-01-06

**Summary:**

Reviewers' concerns are mainly

- Insufficient experimental evidence: Evaluation uses only 10% of Audio-FLAN with LoRA fine-tuning on a single, with outdated model (Qwen2-Audio), without full-data training, scaling studies, or validation across stronger or multiple models.

- Generation claims are weakly supported: Despite claiming unified understanding and generation, the paper provides little convincing evaluation for music and general audio generation, with missing baselines, metrics, and comparisons.

- Limited analysis of the dataset itself: The paper lacks quantitative analysis of instruction diversity, task difficulty, and data quality, and relies heavily on LLM-based normalization and evaluation.

- Generalization and hallucination concerns

**Reviewer Concerns:**

The authors have not reponded to the reviewers, so no concern is addressed.

**Reviewer Scores:**

The authors have not reponded to the reviewers, so I assume the reviewers will keep the original rating.

---

### Decision · Program_Chairs · 2026-01-26

Reject